# Flame resistant cotton lines generated by synergistic epistasis in a MAGIC population

**Gregory N. Thyssen**[1]*, **Brian D. Condon**[1], **Doug J. Hinchliffe**[1], **Linghe Zeng**[2],
**Marina Naoumkina**[3], **Johnie N. Jenkins**[4], **Jack C. McCarty**[4], **Ruixiu Sui**[5],
**Crista Madison**[1], **Ping Li**[3], **David D. Fang**[3]

1 Cotton Chemistry and Utilization Research Unit, USDA-ARS, New Orleans, LA, United States of America,
2 Crop Genetics Research Unit, USDA-ARS, Stoneville, MS, United States of America, 3 Cotton Fiber
Bioscience Research Unit, USDA-ARS, New Orleans, LA, United States of America, 4 Genetics and
Sustainable Agriculture Research Unit, USDA-ARS, Mississippi State, MS, United States of America,
5 Sustainable Water Management Research Unit, USDA-ARS, Stoneville, MS, United States of America

* gregory.thyssen@usda.gov

**Data Availability Statement:** All relevant data are within the manuscript and its Supporting Information files.

## Abstract

Textiles made from cotton fibers are flammable and thus often include flame retardant additives for consumer safety. Transgressive segregation in multi-parent populations facilitates new combinations of alleles of genes and can result in traits that are superior to those of any of the parents. A screen of 257 recombinant inbred lines from a multi-parent advanced generation intercross (MAGIC) population for naturally enhance flame retardance (FR) was conducted. All eleven parents, like all conventional white fiber cotton cultivars produce flammable fabric. MAGIC recombinant inbred lines (RILs) that produced fibers with significantly lower heat release capacities (HRC) as measured by microscale combustion calorimetry (MCC) were identified and the stability of the phenotypes of the outliers were confirmed when the RILs were grown at an additional location. Of the textiles fabricated from the five superior RILs, four exhibited the novel characteristic of inherent flame resistance. When exposed to open flame by standard 45˚ incline flammability testing, these four fabrics self-extinguished. To determine the genetic architecture of this novel trait, linkage, epistatic and multi-locus genome wide association studies (GWAS) were conducted with 473k SNPs identified by whole genome sequencing (WGS). Transcriptomes of developing fiber cells from select RILs were sequenced (RNAseq). Together, these data provide insight into the genetic mechanism of the unexpected emergence of flame-resistant cotton by transgressive segregation in a breeding program. The incorporation of this trait into global cotton germplasm by breeding has the potential to greatly reduce the costs and impacts of flame-retardant chemicals.

## Introduction

Flammability of textiles is an obvious safety and economic concern for textile and cotton consumers, producers and regulatory agencies. Modernizing flame retardant (FR) chemicals has been the focus of research for many years. In recent years, chemists have made numerous advancements in the development of FR treatments for cotton fiber textiles, reducing potential toxicity to end-users and the environment, while also increasing the efficacy of FR [1–9].

**Funding:** The authors received no specific funding for this work.

**Competing interests:** The authors have declared that no competing interests exist.

Some cultivars of cotton (*Gossypium hirsutum* L.) with brown colored fibers have been shown to have intrinsic FR properties [10–13]. Non-woven textiles made from *Lc1* brown cotton fibers self-extinguish upon exposure to an open flame. This cultivar owes both its FR and brown fiber color to higher levels of various flavonoid compounds, due to the activation of an ortholog of the transcription factor *TT2* by a structural variant in the promoter of the gene *Gh_A07G2341*. Cotton fiber cell development proceeds through a series of stages that is measured by days post anthesis (DPA) [14]. The study that identified the FR phenotype also found that the enhanced FR of the fibers was present in developing fibers before the appearance of the brown fiber color, suggesting that a colorless flavonoid or other natural colorless compound was responsible for the FR, rather than the pigment itself [11].

Cotton breeding typically focuses on fiber and yield traits. Each bale of cotton produced in the United States is sampled and tested by a high-volume instrument (HVI). The HVI measures several fiber properties including yellowness (+b), elongation (ELO) which is a measure of how far a fiber can stretch before breaking, micronaire (MIC) which is related to fineness, strength (STR), upper half mean length (UHM), and length uniformity index (UI). These values determine if the bale will be used in low or high value applications, ranging from mops and wipes to high thread count sheets and luxury fashion textiles. Since these HVI measurements are used to set the price of each bale, improvement of these traits is the target of many growers and cotton breeders.

Yield traits in cotton include biotic and abiotic stress tolerance. Genetic markers and candidate genes responsible for herbicide tolerance, pest resistance including nematodes, bacteria and fungi, as well as salt and drought tolerance have been identified and are used for marker assisted selection of advanced and improved cotton breeding materials and commercial cultivars [15–23].

Multi-parent advanced generation intercross (MAGIC) populations are valuable to genetic researchers and breeders and have been developed in many model and commercial crops, including cotton. MAGIC populations create an opportunity for beneficial alleles from the multiple parents to combine in novel ways, resulting in phenotypes that are far superior to any of the parents [24–28]. Transgressive segregation has been observed in simpler, bi-parental populations, but MAGIC populations intrinsically present even greater opportunities.

Here, a MAGIC population that derives from eleven white fiber cotton cultivars was screened for naturally enhanced FR. First, a small-scale experiment, microscale combustion calorimetry (MCC) was used to quantify the heat release capacity (HRC) of the MAGIC lines, a standard proxy for FR [29, 30]. These 257 recombinant inbred lines (RILs) were whole genome sequenced (WGS) at the DNA level, as were the parents, to identify loci that contribute to the novel FR trait. Surprisingly, when fibers from the five RILs with the lowest HRC were used to fabricate non-woven cotton textiles, four white fiber cotton RILs were identified that produce self-extinguishing fabrics. Using multi-locus genome wide association analysis and epistatic interaction analysis, candidate genomic loci and gene variants that are associated with the novel FR phenotype were identified. RNAseq analysis of developing fibers of several of the lines was further used to find candidate genes that are significantly differentially expressed in the naturally self-extinguishing cotton fibers.

## Materials and methods

### Plant materials

The cotton MAGIC population that was used in this study was described previously [31–33]. Briefly, eleven *G. hirsutum* lines representing a broad spectrum of genetic diversity in the USA cotton germplasm were used to create 55 diallel crosses. The resulting lines were subjected to

five cycles of pollen-pooled random mating followed by six generations of self-pollination via single seed descent resulting in 550 RILs. Two hundred fifty-seven of these lines were grown at Stoneville, MS, USA (STV) in 2015 with two replicates and fibers were used for MCC analysis. Then, the thirty highest and thirty lowest HRC lines, selected from the first screen, were grown at Florence, SC, USA (FLO) in 2016 and were also evaluated by MCC to determine the durability of the genetic component of the novel FR trait. The five durably highest and five durably lowest HRC lines were selected based on MCC of fibers from the two locations (STV 2015 and FLO 2016) and planted at STV in 2017. Each of these selected lines was planted in 6 plots of 12.2 × 1.0 m with 60 to 80 plants each plot. The soil type was a Beulah fine sandy loam. Field management followed standard procedures at Mississippi State University Extension Center at Stoneville, MS, USA. The cotton plants were harvested by a two-row mechanical picker and seed cotton from individual lines was collected manually at the chute during the harvesting process. Seed cotton of individual plots was ginned by a microgin equipped with a trash cleaner at the USDA-ARS Ginning Laboratory at Stoneville, MS, USA. The ginned fibers were used to fabricate non-woven textiles for flammability testing, described below.

## Genotyping

The WGS of the MAGIC RILs was described previously [28]. Briefly, 20× coverage of the eleven parents and 3× coverage of the 257 RILs was obtained from Illumina sequencing with paired 100-150bp reads and aligned to the NBIv1 cotton reference genome [34]. After filtering based on quality and KNN interpolation of missing data using TASSEL software, a total of 473,517 SNPs were selected for GWAS, epistatic, and linkage analyses [35].

## Measurement of heat release capacity (HRC)

Two biological replicates of fiber samples from 257 RILs from one location (STV 2015) and 60 outlier RILs replicated at another location (FLO 2016) were subjected to MCC analysis following the standard test method ASTM D7309-13 (ASTM International, West Conshohocken, PA, USA). Each sample was ground to a powder using a Wiley Min-Mill (Thomas Scientific, Swedesboro, NJ, USA) with a 40-mesh sieve. Next ~4 mg of each sample was loaded into a microscale combustion calorimeter model MCC-2 (Deatak, St. McHenry, IL, USA) which used the sample mass and consumption of oxygen during pyrolysis to calculate combustion properties that included heat release capacity (HRC: $J\,g^{-1}\,K^{-1}$), peak heat release rate (pHRR: $W\,g^{-1}$), total heat release (tHR: $kJ\,g^{-1}$), temperature at pHRR (°C), and % mass of sample not consumed (% char yield) as previously described [11]. Raw HRC data were normalized across biological replicates using a best linear unbiased predictor (BLUP) implemented in R software using the lme4 package to fit the model: "model = lmer(phenotype ~ (1|line) + (1|line: replicate))" [36].

## Production of textiles and flammability testing

Cotton fibers from each RIL were chute fed through a Saco Lowell nonwovens card fitted with Cardmaster plates (John D. Hollingsworth on Wheels, Inc., Greenville, SC, USA). The fiber web produced by the card was fed directly into a crosslapper (Technoplants srl., Pistoia, Italy). The number of crosslapps was varied for fibers from each RIL to produce a fabric basis weight of ~100 $g\,m^{-2}$. To produce needle-punched (NP) fabrics, the crosslapped webs were fed continuously into a needleloom (Technoplants srl.) with 3 barb, 9 cm conical needles (Groz-Beckert KG, Albstadt, Germany) and needling impact on the crosslapped web was 130 points $cm^{-2}$ and 490 strokes $min^{-1}$. The produced NP fabrics were subjected to standard flammability testing for apparel using a 45° incline flammability tester (Model TC 45, Govmark Ltd., McHenry, IL,

USA) according to the standard test method ASTM D1230-17 (ASTM International, West Conshohocken, PA, USA).

### Genome wide association analysis

Since transgressive segregation was obvious from the novelty of the FR trait, mrMLM, a multi-locus GWAS software was selected to identify potentially causative loci [37]. This software was run with the 257 RIL HRC data and the 473,517 SNPs in the NBIv1 reference genome using default parameters [28, 34].

### Epistatic analysis

To identify loci that may have interacted with each other to generate the novel FR trait, PLINK software ("-epistasis") was used to analyze epistatic interactions among genetic loci, using default parameters and the same set of 257 RILs and SNPs described above [38].

### Linkage analysis

The WGS of the four RILs with the novel natural self-extinguishing property of their fabricated textiles were also analyzed to identify, by simple linkage, haplotype regions with variants that were held in common among them, but were absent from the four HRC lines with the lowest natural flame retardance that were used to produce textiles and were also subjected to the fabric flammability test. Analysis of the whole genome sequencing was used to identify these common haplotypes by comparing variant call files (VCF), that were generated by samtools and bcftools mpileup software, with bedtools intersect software [39].

### RNA sample collection and transcriptome analysis (RNAseq)

About five plants from each RIL were grown during the summer of 2019 in a field in New Orleans, LA, USA. Cotton bolls were collected from each plant at 8- and 16-days post anthesis (DPA). Cotton fibers were separated from ovules of each individual plant and pooled into two samples comprising two replicates per RIL. Total RNA was isolated from detached fibers using the Sigma Spectrum Plant Total RNA Kit (Sigma-Aldrich, St. Louis, MO) with the optional on column DNase1 digestion according to the manufacturer's protocol. RNA samples were paired-end Illumina (Platform PE150) sequenced with over 20 million paired raw reads per sample by Novogene Corporation (Chula Vista, CA, USA). The paired 150 bp reads from each of eight MAGIC RILs were analyzed by alignment to the NBIv1 reference genome with HISAT2 software and reads per kilobase per million reads (RPKM) in annotated genes were counted using samtools and bedtools software [34, 39–41]. Differentially expressed genes were identified based on the Student's t-test globally and in select regions of chromosomes that were identified by the GWAS, epistatic and simple linkage analyses. Genes with >1 RPKM in at least one sample were considered to be significantly differentially expressed if the average expression in the four FR lines was at least 2-fold different from the flammable lines, and that the p-value of this comparison was <0.1.

## Results

### Transgressive segregation of heat release capacity (HRC) in MAGIC RILs

The distribution of HRC values obtained by screening the 257 MAGIC RILs extended well beyond the range of values observed in the eleven parents (Fig 1A, S1 Table) indicating transgressive segregation to generate the unexpected FR trait. Since FR has been previously observed in some naturally brown colored cotton fibers, which are also known to have

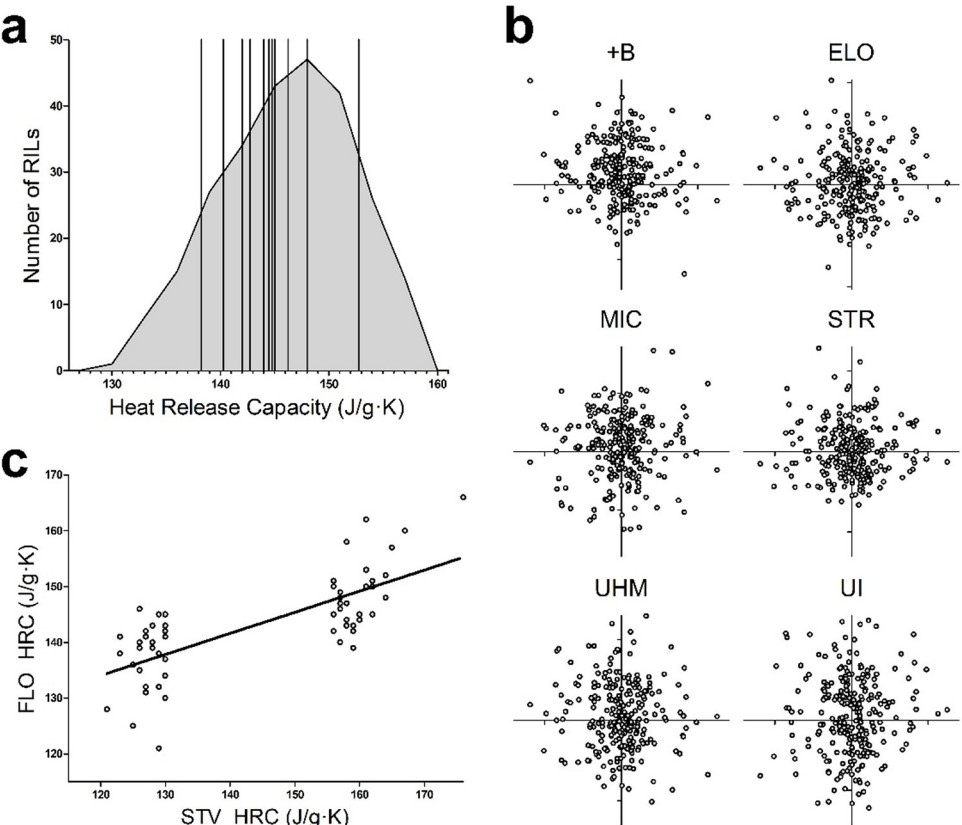

**Fig 1. Heat release capacity (HRC) is a phenotype of cotton fibers measured by micro-scale combustion calorimetry.** Low HRC correlates with enhanced flame resistance. (a) Histogram of HRC for 257 RILs reveals transgressive segregation. Vertical lines show the HRC values for the 11 parents of the MAGIC population. (b) HRC (x-axis) is not correlated with other fiber traits. (+B, yellowness; ELO, elongation; MIC, micronaire; STR, strength; UHM, upper half mean length; UI, uniformity index). (c) Stability of HRC at two growing locations. The screen of fibers grown at Stoneville, MS, USA (STV, x-axis) was used to select 60 outlier RILs that were grown in Florence, SC, USA (FLO, y-axis).

generally inferior HVI fiber traits, correlation between the HRC values and +b, ELO, MIC, STR, UHM, UI was investigated, but no significant correlation was found (Fig 1B). To determine the stability and durability of the novel FR trait, the 30 highest and 30 lowest HRC lines were selected based on the Stoneville, MS, USA (STV) data from 2015, and then fibers from those RILs grown in a different location, Florence, SC, USA (FLO) in 2016, were tested by MCC, showing a correlation that supports the genetic basis of the trait (Fig 1C).

## Production of naturally self-extinguishing non-woven textiles

The five lines with the highest and the five lines with the lowest small-scale HRC values, as validated at both locations (STV 2015, FLO 2016), were grown for a third time (STV 2017), to collect cotton fibers for textile production. Non-woven textiles from each of these ten lines were fabricated and the fabrics were subjected to standard 45˚ incline flame testing. Textiles from four of the five lines with the lowest HRC self-extinguished in the five technical replicates of the experiment. The other six lines were rapidly and completely consumed by flame (Figs 2 and 3, S1 and S2 Movies).

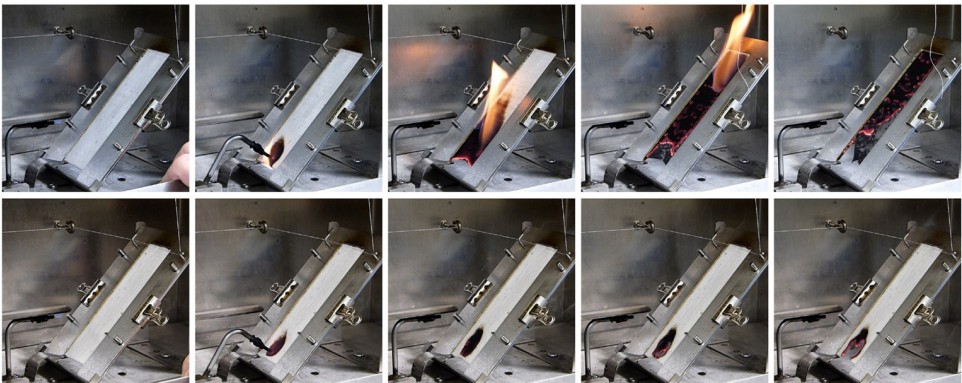

**Fig 2. Time series of 45˚ incline flame test of non-woven fabrics from the MAGIC RILs with the inferior HRC and the most superior HRC.** Each image is 5 seconds apart. Top series is fabric made from RIL-225, which like all untreated textiles produced from conventional cultivated white cottons, was fully consumed by flame in ~15s. Bottom series is RIL-385, which self-extinguished. See also Fig 3 and S1 and S2 Movies.

## Multi-locus GWAS for HRC

The HRC data for the tested 257 RILs along with the WGS genotype data of 473,517 SNPs were analyzed with mrMLM software, a multi-locus GWAS analysis. Thirteen highly significant loci on 11 chromosomes were identified (Fig 4, S2 Table).

## Genome wide epistasis study

PLINK software was used to identify pairs of loci that influence the HRC values of the 257 RILs tested. Significant, predicted epistatic interactions were linked to 26 loci. Fig 4 presents one network of the most highly significant epistatic loci in red, a second larger network in blue and the remaining highly significant epistatic loci in green, in the center of the Circos plot. The specific, significant SNP pairs identified by PLINK are presented in S3 Table.

## Linkage analysis

Although fabrics from only ten lines were generated, the genomic loci that are held in common among the four lines with the novel trait of self-extinguishing white cotton fiber and absent from the flammable lines were identified using the WGS data. These are presented by the black lines in Fig 4 and the SNP locations are listed in S4 Table. Several of these loci are within a few Mb of loci also identified by PLINK and/or mrMLM, mentioned above, but none exactly match (Fig 4, S2–S4 Tables).

## Transcriptome analysis

RNAseq data was collected for eight RILs that were made into fabrics and subjected to flame testing, including the four lines that produced self-extinguishing fabrics, and four that did not. Relative to the flammable lines, 266 genes were down-regulated and 50 were up-regulated in FR lines at 8-DPA (S6 and S7 Tables). Relative to the flammable lines, 90 genes were down-regulated and 216 were up-regulated in FR lines at 16-DPA (S8 and S9 Tables). Among the 602 unique genes, 41 segregated for non-synonymous variants in the RMUP-MAGIC population (S6–S9 Tables). Only 13 of these highly differentially expressed genes were within 100-kb of the locations identified by mrMLM, PLINK or linkage analysis (Table 1, S10 Table).

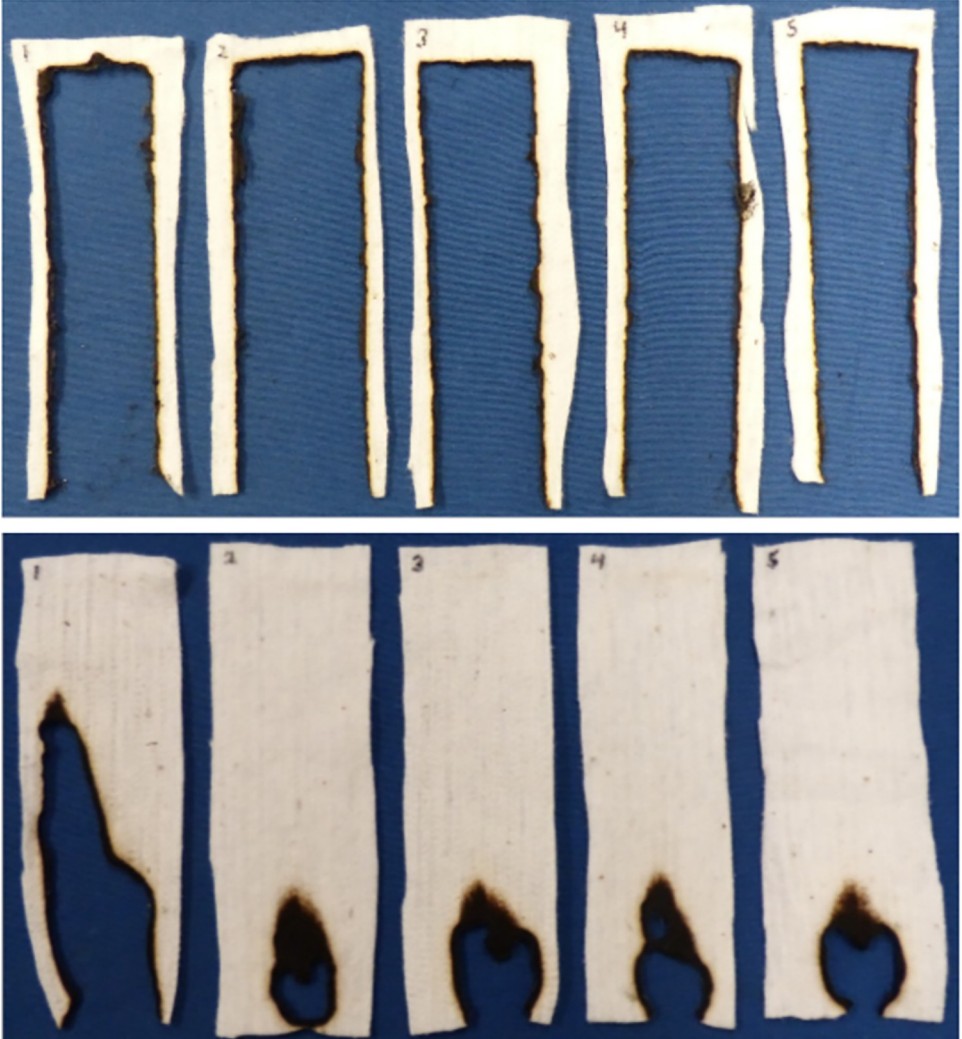

**Fig 3. End point of five technical replicates of the 45˚ incline flame test of non-woven fabrics from the MAGIC RILs with the inferior HRC and the most superior HRC.** Top series is fabric made from RIL-225, which like all untreated textiles produced from conventional cultivated white cottons, was fully consumed by flame in ~15s. Bottom series is RIL-385, which self-extinguished. See also Fig 2 and S1 and S2 Movies.

## Discussion

The discovery of white cotton fibers that can make fabrics with intrinsic self-extinguishing flame-retardant properties is surprising and important for the efficient development of safe textiles. These RILs, and other yet to be identified white FR cultivars, should be incorporated into cotton breeding programs even though, at this time, the genetic basis is perhaps not clearly enough defined for marker assisted selection, which necessitates the use of MCC phenotyping in breeding programs. Here, a novel phenotype of natural flame retardance in white cotton MAGIC RILs has been shown to be based on genetics and transgressive segregation.

The novel FR trait is obviously due to an uncommon combination of alleles, since none of the eleven parents possess the trait, so a multi-locus GWAS methodology was selected. The mrMLM software identified multiple loci that are associated with the FR trait. Since no SNP in this population had more than two alleles, we did not employ a multi-allelic model, although

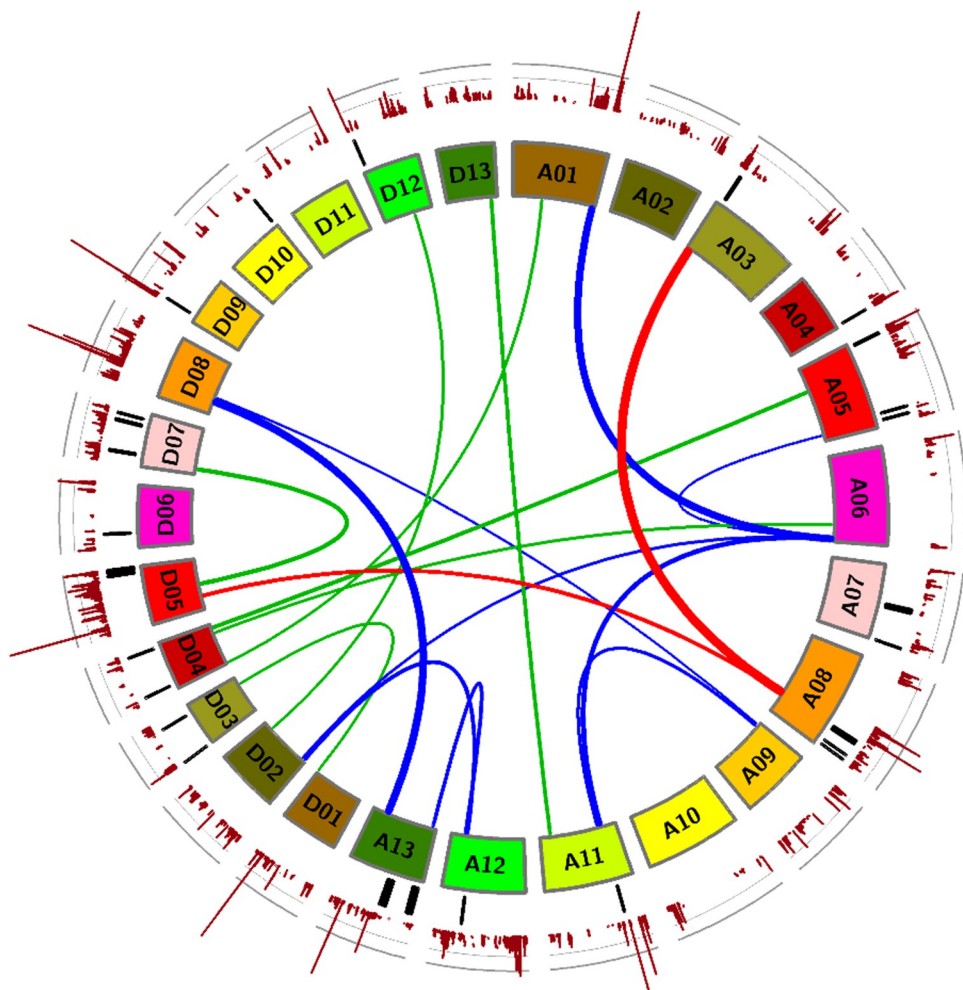

**Fig 4. GWAS loci, epistatic interactions and genetically linked loci influence flammability of cotton fibers.** The thickness of the innermost links between chromosomes reflect p-values of the 242 most highly significant (-logP > 9) epistatic interactions detected by PLINK, using 473,517 SNPs and HRC data from 257 RILs. The colors of these links facilitate identification of potential multi-gene networks. The outer-most plot represents mrMLM GWAS results. Black lines in between indicate the 405 SNPs that are shared by the RILs that made self-extinguishing fabrics and are absent from the RILs that did not. See also S2–S4 Tables.

future analysis using haplotype bins may be appropriate for future work. Here, we further used PLINK software to identify loci that may work in concert to orchestrate the novel FR trait by synergistic epistasis. Also, a simplistic haplotype analysis identified chromosomal regions that are held in common among the four lines with promising HRC and which generated self-extinguishing fabrics but were absent from the lines with the highest HRC values that generated flammable textiles. These analyses were used to select the candidate loci and genes that were further investigated by RNAseq.

As mentioned above, the candidate chromosomal loci and genes are so far preliminary. However, there were multiple differentially expressed genes near the candidate loci that deserve further investigation. One especially interesting candidate, *Gh_A13G0444*, is part of the flavonoid biosynthetic pathway that has been previously suggested to be the source of inherent FR properties in brown fiber cotton lines [11, 12]. This gene is predicted to conjugate 2-oxogluterate with the colorless flavonoid leucoanthocyanidin to generate the 4-H-

**Table 1. RNAseq expression of select genes.** Annotated genes with at least 2-fold differential expression between FR and flammable lines and were located within 100-kb of a chromosomal location identified in the genetic analysis are shown. Expression is shown in RPKM units, and BLUP-normalized units are shown for heat release capacity (HRC), which represent the deviation from the mean in J $g^{-1}$ $K^{-1}$. See also S10 Table for these gene descriptions and S6–S9 Tables for additional differentially expressed genes, and S5 Table for expression of all genes expressed >0 RPKM.

| GhNBI gene | | Gh_A03G0325 | Gh_A05G0191 | Gh_A05G0200 | Gh_A07G1441 | Gh_A08G1100 | Gh_A08G1149 | Gh_A11G0806 | Gh_A12G1044 | Gh_A13G0444 | Gh_D05G3710 | Gh_D07G2233 | Gh_D11G3237 | Gh_D11G3253 | HRC BLUP (J $g^{-1}$ $K^{-1}$) |
|---|---|---|---|---|---|---|---|---|---|---|---|---|---|---|---|
| NonSyn | | | | | | | | | | | | X | X | X | |
| mrMLM | | | X | X | | | | X | X | X | | X | X | X | |
| PLINK | | | | | | X | | | | | | | | | |
| Linkage | | X | | | X | | X | | X | X | X | | | | |
| 8-DPA p-val | | 0.038 | 0.372 | 0.092 | 0.225 | 0.004 | 0.414 | 0.063 | 0.310 | 0.029 | 0.044 | 0.964 | 0.758 | 0.433 | |
| 16-DPA p-val | | 0.070 | 0.051 | 0.647 | 0.009 | 0.033 | 0.018 | 0.093 | 0.020 | 0.044 | 0.900 | 0.041 | 0.057 | 0.023 | |
| 8-DPA -log2Fold | | 0.95 | -2.55 | -1.11 | -0.57 | -0.51 | -0.16 | 0.67 | 0.24 | -1.41 | -1.99 | -0.21 | 0.03 | 0.36 | |
| 16-DPA -log2Fold | | 1.46 | 1.14 | -0.45 | -1.41 | 1.17 | 1.38 | 1.58 | 2.63 | -1.31 | -0.15 | -1.28 | 1.03 | 1.14 | |
| DPA | RIL | | | | | | | | | | | | | | |
| 8 | 385 | 62.3 | 0.5 | 0.3 | 1.1 | 2.6 | 1.5 | 3.6 | 19.6 | 2.2 | 1.0 | 0.4 | 8.0 | 33.6 | -5.94 |
| 8 | 532 | 46.2 | 0.5 | 0.3 | 1.4 | 2.8 | 1.6 | 3.5 | 16.7 | 1.3 | 0.7 | 0.3 | 4.9 | 31.9 | -5.05 |
| 8 | 505 | 58.3 | 0.5 | 0.3 | 1.6 | 1.9 | 2.3 | 4.1 | 20.7 | 3.0 | 0.6 | 0.1 | 7.7 | 25.4 | -4.28 |
| 8 | 052 | 118.1 | 1.1 | 0.4 | 1.5 | 1.9 | 2.5 | 2.5 | 23.7 | 1.2 | 0.3 | 0.5 | 6.6 | 26.5 | -4.22 |
| 8 | 273 | 44.2 | 0.9 | 1.2 | 2.1 | 3.5 | 2.0 | 1.9 | 15.6 | 3.8 | 2.4 | 0.5 | 4.4 | 37.1 | 4.23 |
| 8 | 531 | 35.2 | 13.5 | 0.6 | 3.5 | 3.5 | 1.9 | 2.9 | 16.3 | 3.9 | 4.9 | 0.3 | 4.5 | 11.6 | 4.91 |
| 8 | 375 | 36.4 | 0.4 | 0.7 | 0.8 | 2.9 | 3.0 | 1.8 | 12.7 | 3.8 | 0.8 | 0.4 | 6.6 | 30.7 | 5.66 |
| 8 | 225 | 31.3 | 0.4 | 0.3 | 1.9 | 3.2 | 1.9 | 2.0 | 23.7 | 9.0 | 2.2 | 0.3 | 11.1 | 12.1 | 6.26 |
| 16 | 385 | 1.5 | 2.3 | 0.3 | 5.2 | 2.3 | 1.7 | 0.9 | 8.8 | 1.8 | 1.3 | 0.1 | 47.3 | 30.5 | -5.94 |
| 16 | 532 | 0.3 | 1.3 | 0.9 | 29.3 | 1.7 | 1.4 | 0.3 | 3.4 | 0.8 | 2.8 | 0.4 | 17.7 | 43.4 | -5.05 |
| 16 | 505 | 0.5 | 0.8 | 0.2 | 16.2 | 1.0 | 0.9 | 0.2 | 2.0 | 2.3 | 0.4 | 0.0 | 24.0 | 22.1 | -4.28 |
| 16 | 052 | 1.0 | 1.1 | 0.8 | 11.0 | 1.3 | 1.2 | 1.3 | 4.4 | 2.3 | 8.3 | 0.2 | 22.2 | 34.7 | -4.22 |
| 16 | 273 | 0.4 | 0.4 | 1.1 | 55.7 | 0.4 | 0.8 | 0.5 | 0.6 | 4.2 | 2.5 | 0.4 | 16.2 | 20.7 | 4.23 |
| 16 | 531 | 0.0 | 0.4 | 0.4 | 41.1 | 1.4 | 0.5 | 0.1 | 1.4 | 3.5 | 1.1 | 0.5 | 18.0 | 6.0 | 4.91 |
| 16 | 375 | 0.5 | 1.1 | 0.9 | 23.2 | 0.5 | 0.4 | 0.1 | 0.2 | 3.1 | 4.3 | 0.3 | 8.3 | 23.4 | 5.66 |
| 16 | 225 | 0.3 | 0.6 | 0.6 | 44.3 | 0.5 | 0.3 | 0.2 | 0.8 | 7.0 | 6.3 | 0.5 | 11.8 | 9.2 | 6.26 |

anthocyanidin pigment (KEGG ortholog E1.14.11.19; leucoanthocyanidin dioxygenase, KEGG pathways ko01100, ko01110, ko00941, K05277) [42]. *Gh_A13G0444* contains a nonsynonymous mutation that is homozygous in all four of the lines that generated FR fabrics. It is ~2.5-fold under-expressed at both 8- and 16-DPA in the FR lines, suggesting the possibility of restricted flux from leucoanthocyanidin to anthocyanidin, and the accumulation of colorless flavonoids that share the type of ring structure that has been proposed to contribute to the FR properties of tannins [5]. However, due to the preliminary nature of the genomic analysis presented here, further work including isolation of metabolites from developing fibers will be necessary to test this hypothesis.

Another interesting candidate gene is *Gh_A12G1044*, which is predicted to be involved in wax biosynthesis (KEGG ortholog EC:2.3.1.180; fabH 3-oxoacyl-synthase III, KEGG pathways ko01100, ko01212, ko00061, K00648) [42]. It also contains a nonsynonymous mutation in the FR lines relative to the reference genome sequence. It is ~6-fold over-expressed at 16-DPA in the FR lines that generated self-extinguishing fabrics.

However, neither of these genes alone can explain the novel FR trait, as these alleles were already present in some of the parent lines. Due to the novelty of this FR trait, and the absence of fiber-like seed trichomes in model organisms like *Arabidopsis*, which was used for annotations of gene function, it is not yet possible to conclusively define the critical genes. However, this analysis of the genetic architecture, gene expression and metabolomes of cotton plants that exhibit the novel fiber FR trait will, with significant further study, lead to opportunities for marker assisted selection for efficient introgression of inherent flame resistance into cultivated cotton.

## Conclusion

Textiles made from cotton fibers are flammable and thus often include flame retardant additives for consumer safety. Breeding programs that combine multiple parents can create new combinations of variants of genes that results in traits that are superior to those of any of the parents. All eleven MAGIC parents, like other conventional white fiber cotton cultivars, produce flammable fabrics. The new cotton lines generated textiles with the novel characteristic of inherent flame resistance (FR). When exposed to open flame by standard flammability testing procedures, textiles made from these lines self-extinguished. Linkage, epistatic, transcriptomic and multi-locus genome wide association studies (GWAS) from multiple locations and years suggest that heritable, genetically controlled synergistic epistasis created the novel trait by a combination of numerous small-effect alleles and genetic loci. Mapping such small effect, synergistic loci is difficult, and much further research is required to fully understand the mechanism of the natural flame retardance. However, these novel lines can be used as germplasm in breeding programs since phenotypic selection on a population, based on micro-scale combustion calorimetry, is feasible, as shown in this study. Breeding of inherently flame-resistant white cotton varieties has the potential to reduce the costs and impacts of use of flame-retardant chemicals, and benefit textile producers and consumers.

## Supporting information

**S1 Movie. 45˚ incline textile flammability test of RIL-225.**
(MP4)

**S2 Movie. 45˚ incline textile flammability test of RIL-385.**
(MP4)

**S1 Table. Heat release capacity of eleven parents and the RILs that were used for RNAseq and fabric flame testing.** See also Fig 1A.
(XLSX)

**S2 Table. Highly significant loci identified by mrMLM multi-locus GWAS analysis.**
(XLSX)

**S3 Table. List of significant epistatic pairs identified by PLINK.**
(XLSX)

**S4 Table. List of SNPs that were present in the four FR lines that produced self-extinguishing fabric, but absent from the most flammable RILs.**
(XLSX)

**S5 Table. RNAseq expression (RPKM) of all annotated genes with >0 RPKM from the selected lines at 8-DPA and 16-DPA (N = 41,746).**
(XLSX)

**S6 Table. RNAseq expression (RPKM) of genes that were 2-fold less expressed in FR lines at 8-DPA.**
(XLSX)

**S7 Table. RNAseq expression (RPKM) of genes that were 2-fold more expressed in FR lines at 8-DPA.**
(XLSX)

**S8 Table. RNAseq expression (RPKM) of genes that were 2-fold less expressed in FR lines at 16-DPA.**
(XLSX)

**S9 Table. RNAseq expression (RPKM) of genes that were 2-fold more expressed in FR lines at 16-DPA.**
(XLSX)

**S10 Table. Expression of select genes, as in Table 1, plus Arabidopsis orthologs and gene descriptions.**
(XLSX)

## Acknowledgments

We thank Dr. Todd Campbell in Florence, SC, USA, for conducting part of field experiments and Mr. Chris Delhom and Mrs. Holly King in New Orleans, LA, USA, for measuring the fiber quality attributes. Mention of trade names or commercial products in this article is solely for the purpose of providing specific information and does not imply recommendation or endorsement by the USDA which is an equal opportunity provider and employer.

## Author Contributions

**Conceptualization:** Gregory N. Thyssen, Brian D. Condon, Doug J. Hinchliffe, Johnie N. Jenkins, Jack C. McCarty.

**Data curation:** Gregory N. Thyssen.

**Formal analysis:** Gregory N. Thyssen, Doug J. Hinchliffe, David D. Fang.

**Investigation:** Gregory N. Thyssen, Doug J. Hinchliffe, Marina Naoumkina, Johnie N. Jenkins, Jack C. McCarty, Crista Madison, Ping Li.

**Methodology:** Gregory N. Thyssen, Doug J. Hinchliffe, Linghe Zeng, Marina Naoumkina, Johnie N. Jenkins, Jack C. McCarty, Ruixiu Sui, Crista Madison, Ping Li, David D. Fang.

**Project administration:** Brian D. Condon, Jack C. McCarty.

**Resources:** Brian D. Condon, Doug J. Hinchliffe, Linghe Zeng, Marina Naoumkina, Johnie N. Jenkins, Jack C. McCarty, Ruixiu Sui, David D. Fang.

**Supervision:** Brian D. Condon, David D. Fang.

**Writing – original draft:** Gregory N. Thyssen.

**Writing – review & editing:** Brian D. Condon, Doug J. Hinchliffe, Linghe Zeng, Marina Naoumkina, Johnie N. Jenkins, Jack C. McCarty, Ruixiu Sui, Crista Madison, Ping Li, David D. Fang.

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
