## [Decision Letter · Decision Letter 0]

8 Aug 2022

PONE-D-22-19253Flame resistant cotton lines generated by synergistic epistasis in a MAGIC populationPLOS ONE

Dear Dr. Thyssen,

Thank you for submitting your manuscript to PLOS ONE. After careful consideration, we feel that it has merit but does not fully meet PLOS ONE’s publication criteria as it currently stands. Therefore, we invite you to submit a revised version of the manuscript that addresses the points raised during the review process. Please submit your revised manuscript by Sep 22 2022 11:59PM. If you will need more time than this to complete your revisions, please reply to this message or contact the journal office at plosone@plos.org. Please include the following items when submitting your revised manuscript:A rebuttal letter that responds to each point raised by the academic editor and reviewer(s). You should upload this letter as a separate file labeled 'Response to Reviewers'.A marked-up copy of your manuscript that highlights changes made to the original version. You should upload this as a separate file labeled 'Revised Manuscript with Track Changes'.An unmarked version of your revised paper without tracked changes. You should upload this as a separate file labeled 'Manuscript'.

We look forward to receiving your revised manuscript.

Kind regards,

Yuan-Ming Zhang

Academic Editor

PLOS ONE

Journal Requirements:

2. Please amend your list of authors on the manuscript to ensure that each author is linked to an affiliation. Authors’ affiliations should reflect the institution where the work was done (if authors moved subsequently, you can also list the new affiliation stating “current affiliation:….” as necessary).

Additional Editor Comments:

I have received two review reports. The two reviewers gave some comments and suggestions. Please address them carefully in the revision.

Reviewers' comments:

Reviewer's Responses to Questions

**Comments to the Author**

1. Is the manuscript technically sound, and do the data support the conclusions?

Reviewer #1: Yes

Reviewer #2: Yes

2. Has the statistical analysis been performed appropriately and rigorously? 

Reviewer #1: No

Reviewer #2: No

3. Have the authors made all data underlying the findings in their manuscript fully available?

Reviewer #1: Yes

Reviewer #2: Yes

4. Is the manuscript presented in an intelligible fashion and written in standard English?

Reviewer #1: Yes

Reviewer #2: Yes

5. Review Comments to the Author

Reviewer #1: In this manuscript entitled “Flame resistant cotton lines generated by synergistic epistasis in a MAGIC population” presented the identification of the lower heat release capacities textiles made from cotton fiber produced from MAGIC RILs. The flame-resistant trait of cotton germplasm has the potential to greatly reduce the costs of flame retardant additives for cotton textiles. To determine the genetic mechanism of the flame-resistant cotton by transgressive segregation, the authors used the GWAS and RNAseq methods, identified 26 loci and 13 genes were linked to the influence the HRC values in the 257 RILs. The authors showed an interesting topic, however, there are some concerns and suggestions for authors to address:

1. How about the HRC values of the eleven parents of MAGIC? The authors should to show the experiments and results in the MS.

2. Whether there is a difference in the flavonoid content between the flammable lines and FR lines? to confirm the previous suggestion that a colorless flavonoid or other natural colorless compound was responsible for the FR. And to support the GWAS and RNAseq data.

3. The GWAS and RNAseq results are too preliminary, it does not make a whole story.

4. To further focus on the 41 segregated for non-synonymous variants in the RMUP-MAGIC population among the 602 unique genes, and the 13 highly differentially expression genes within the locations, to determine the genetic mechanism of the flame-resistant cotton.

Reviewer #2: The natural flame resistant is an important trait in cotton. Linkage, epistatic, GWAS and transcriptomes analysis were conducted to determine the genetic architecture in MAGIC population of cotton. Of the textiles fabricated from the five offspring of MAGIC, four exhibited the novel characteristic of inherent flame resistance. It is expected to provide guidance for breeding. This is an interesting study. But I have some concerns in statistical analysis.

1. The greatest advantages of QTL mapping using MAGIC populations come from multiple founder alleles. Many researches think that multi-allele model should be considered in the statistical analysis of MAGIC population. Multi-allele model can estimate the effect of every allele and it is beneficial to crop breeding. In this study the GWAS method was used to analyze the datasets of 11-parent MAGIC population, but all GWAS methods consider only two alleles. Please explain why the multi-allele model did not use.

2. When the GWAS method was used to analyze the dataset of MAGIC population, please consider whether the population structure should be considered.

These reasons may make your results difficult to understand. For example, the loci identified in linkage analysis are apart from the loci identified by PLINK and/or mrMLM.

6. PLOS authors have the option to publish the peer review history of their article (what does this mean?). If published, this will include your full peer review and any attached files.

Reviewer #1: No

Reviewer #2: No

---

## [Author Response · Author response to Decision Letter 0]

10 Aug 2022

August 10, 2022

Dear Dr. Yuan-Ming Zhang, Reviewers and Editorial Board of PLOS ONE:

I am submitting a revised manuscript for your consideration, titled, “Flame resistant cotton lines generated by synergistic epistasis in a MAGIC population.” [PONE-D-22-19253]

We appreciate your time, consideration, and valuable comments. Below we have responded to each of the comments and have made modifications to the manuscript to reflect your requests and suggestions.

Sincerely, 

Gregory Thyssen, Ph.D.

USDA-ARS

Comments to the Author:

2. Has the statistical analysis been performed appropriately and rigorously? 

Reviewer #1: No

Reviewer #2: No

RESPONSE: 

All p-values shown in the Tables and Supplemental Tables were computed using the default statistical methods implemented by the highly cited bioinformatic software described in the Methods. No attempt was made to artificially reduce thresholds for significance or to employ other-than-default statistical analysis. Rather than presenting just one of the genomic analyses, we present three (PLINK, mrMLM, simple linkage), which do not completely agree with each other. We agree that the genomics part of this report is preliminary, so we believe that presenting all of these data will be the most beneficial to the other researchers and breeders that will hopefully help continue this work, so that naturally flame-resistant cotton can be further characterized and used in breeding programs.

Reviewer #1:

In this manuscript entitled “Flame resistant cotton lines generated by synergistic epistasis in a MAGIC population” presented the identification of the lower heat release capacities textiles made from cotton fiber produced from MAGIC RILs. The flame-resistant trait of cotton germplasm has the potential to greatly reduce the costs of flame retardant additives for cotton textiles. To determine the genetic mechanism of the flame-resistant cotton by transgressive segregation, the authors used the GWAS and RNAseq methods, identified 26 loci and 13 genes were linked to the influence the HRC values in the 257 RILs. The authors showed an interesting topic, however, there are some concerns and suggestions for authors to address:

1. How about the HRC values of the eleven parents of MAGIC? The authors should show the experiments and results in the MS.

RESPONSE:

The HRC values of the eleven parents are represented by the vertical black lines in the histogram in Figure 1a. Now, we have added mention of the parents in the Methods and a new Supplemental Table 1 with the numerical values.

2. Whether there is a difference in the flavonoid content between the flammable lines and FR lines? to confirm the previous suggestion that a colorless flavonoid or other natural colorless compound was responsible for the FR. And to support the GWAS and RNAseq data.

REPSONSE:

Quantification of flavonoid compounds and other metabolites in the flame resistant and flammable lines is ongoing. Unfortunately, due to the COVID pandemic, our collaborator that was working on this has not yet been able to complete the study. We may need to grow another season for fresh material and/or replication of the experiment. Now, we have modified the manuscript to emphasize that the flavonoid mechanism remains at this point a hypothesis for future work and not a claim of a finding. As mentioned above, publication of this work will hopefully encourage other groups to test the material and develop additional samples, hypotheses, and findings.

3. The GWAS and RNAseq results are too preliminary, it does not make a whole story.

RESPONSE:

We agree that the genomics part of this report is preliminary. Rather than work in isolation to discover the complete mechanism, we choose to share the candidate genes and loci that we could find already by various approaches. Since the mechanism is likely the synergistic interaction of numerous small-effect loci, much further work will be required to present the whole story. However, these FR lines already have value to breeders since phenotypic selection by MCC is feasible at scale. Now, we acknowledge and further emphasize these points in the Conclusion. Again, exposure of our findings to other researchers and breeders is the fastest way for the whole story to be discovered.

4. To further focus on the 41 segregated for non-synonymous variants in the RMUP-MAGIC population among the 602 unique genes, and the 13 highly differentially expression genes within the locations, to determine the genetic mechanism of the flame-resistant cotton.

RESPONSE:

We do intend to further focus on our preliminary candidate genes to identify the whole story of the mechanism, and to develop genetic markers for marker assisted breeding. However, this requires another growing season, so that more stages of the developing cotton fiber tissues can be collected for RNA extraction and quantification. We are also currently developing additional informative populations to isolate and study subsets of the preliminary candidate genes. Publication of the present study will hopefully lead to study of these lines and our currently identified FR RILs at additional locations, both by our collaborators and other independent colleagues in cotton research and breeding.

Reviewer #2: 

The natural flame resistant is an important trait in cotton. Linkage, epistatic, GWAS and transcriptomes analysis were conducted to determine the genetic architecture in MAGIC population of cotton. Of the textiles fabricated from the five offspring of MAGIC, four exhibited the novel characteristic of inherent flame resistance. It is expected to provide guidance for breeding. This is an interesting study. But I have some concerns in statistical analysis.

1. The greatest advantages of QTL mapping using MAGIC populations come from multiple founder alleles. Many researchers think that multi-allele model should be considered in the statistical analysis of MAGIC population. Multi-allele model can estimate the effect of every allele and it is beneficial to crop breeding. In this study the GWAS method was used to analyze the datasets of 11-parent MAGIC population, but all GWAS methods consider only two alleles. Please explain why the multi-allele model did not use.

RESPONSE:

Gossypium hirsutum cultivars have limited genetic diversity and a narrow genetic base, especially when compared to other crops like Zea mays. In this population, within the 473,000 high quality SNPs, there are no multi-allelic SNPs. Now, the manuscript has been modified to highlight this fact. Further analysis to explore the utility of a multi-allelic analysis could be achieved by first identifying haplotype blocks or regions of identity-by-descent that are large enough to include enough polymorphic SNPs to define multiple alleles of each haplotype. So far, we have not found such, but we agree with the suggestion and will continue to explore multi-allele models in future work.

2. When the GWAS method was used to analyze the dataset of MAGIC population, please consider whether the population structure should be considered.

These reasons may make your results difficult to understand. For example, the loci identified in linkage analysis are apart from the loci identified by PLINK and/or mrMLM.

RESPONSE:

This structure of this MAGIC population has been characterized several times. First with 1,600 SSR markers, then with 6,000 SNP markers, and again with the 473,000 SNPs identified by whole genome sequencing used in the present study. [Refs 31, 32, 28 in manuscript]. These all indicated that the five cycles of bulked-pollen random mating, followed by six generations of self-pollination, was successful at creating a population with no significant population structure.

We hypothesize that the discrepancies between PLINK and mrMLM are due to the inherent difficulty in identifying the mechanism of a trait that is controlled by numerous small effect loci and synergistic epistatic interactions. Since none of the parent lines possess the flame resistance phenotype, elucidation of the mechanism is not straightforward. We believe that presenting the readers with the multiple genetic analyses, though some loci differ between the approaches, is more useful for the advancement of the research by the larger cotton research community than to stretch to reconcile them, at this point.

---

## [Decision Letter · Decision Letter 1]

24 Oct 2022

PONE-D-22-19253R1Flame resistant cotton lines generated by synergistic epistasis in a MAGIC populationPLOS ONE

Dear Dr. Thyssen,

Thank you for submitting your manuscript to PLOS ONE. After careful consideration, we feel that it has merit but does not fully meet PLOS ONE’s publication criteria as it currently stands. Therefore, we invite you to submit a revised version of the manuscript that addresses the points raised during the review process.

We look forward to receiving your revised manuscript.

Kind regards,

Yuan-Ming Zhang

Academic Editor

PLOS ONE

Journal Requirements:

Additional Editor Comments:

I have received one additional report. Some minor revision should be addressed. Look forward to receiving your further revision soon!

Reviewers' comments:

Reviewer's Responses to Questions

**Comments to the Author**

1. If the authors have adequately addressed your comments raised in a previous round of review and you feel that this manuscript is now acceptable for publication, you may indicate that here to bypass the “Comments to the Author” section, enter your conflict of interest statement in the “Confidential to Editor” section, and submit your "Accept" recommendation.

Reviewer #2: (No Response)

Reviewer #3: (No Response)

2. Is the manuscript technically sound, and do the data support the conclusions?

Reviewer #2: (No Response)

Reviewer #3: Yes

3. Has the statistical analysis been performed appropriately and rigorously? 

Reviewer #2: (No Response)

Reviewer #3: Yes

4. Have the authors made all data underlying the findings in their manuscript fully available?

Reviewer #2: (No Response)

Reviewer #3: Yes

5. Is the manuscript presented in an intelligible fashion and written in standard English?

Reviewer #2: (No Response)

Reviewer #3: Yes

6. Review Comments to the Author

Reviewer #2: (No Response)

Reviewer #3: 1. Line 92: remove ref 31.

2. Line 159-162: can you explain more how you did linkage analyses using 4 RILs? It is an haplotype analyses to compare the 4 lines?

3. Line 249-254: In table S4, could you add those loci within a few Mb from PLINK or mrMLM?

4. Line 256-264: could you describe more about any of those 13 genes are potential candidates or consistent with any other published research? I understand that the flammable traits may not have may published results. Candidate for any traits highly correlated with it? You mentioned some in the discussion but you may want to explain more in the main text.

5. Line 290: using highest or lowest HRC values instead of worst.

6. Line 327: remove “Found among”, and “were lines that”.

7. Line 335: remove “already”.

7. PLOS authors have the option to publish the peer review history of their article (what does this mean?). If published, this will include your full peer review and any attached files.

Reviewer #2: No

Reviewer #3: **Yes: **Shuyu Liu

---

## [Author Response · Author response to Decision Letter 1]

8 Nov 2022

November 7, 2022

Dear Dr. Yuan-Ming Zhang, Reviewers and Editorial Board of PLOS ONE:

I am submitting a twice revised manuscript for your consideration, titled, “Flame resistant cotton lines generated by synergistic epistasis in a MAGIC population.” [PONE-D-22-19253]

We appreciate your time, consideration, and valuable comments. Below we have responded to each of the new comments and have made additional modifications to the manuscript to reflect your requests and suggestions.

Sincerely, 

Gregory Thyssen, Ph.D.

Computational Biologist

Cotton Chemistry and Utilization Unit &

Cotton Fiber Bioscience Unit

USDA-ARS

Reviewer #3: 

1. Line 92: remove ref 31.

>>Done.

2. Line 159-162: can you explain more how you did linkage analyses using 4 RILs? It is an haplotype analyses to compare the 4 lines?

>>Added:

“Analysis of the whole genome sequencing was used to identify these common haplotypes by comparing variant call files (VCF), that were generated by samtools and bcftools mpileup software, with bedtools intersect software [39]."

3. Line 249-254: In table S4, could you add those loci within a few Mb from PLINK or mrMLM?

>>We have added a new Table S5 that includes all the RNAseq expression data for every annotated gene with any detectible expression (>0 RPKM) (N=41,746) in the 8 RILs at developmental stages of 8-DPA and 16-DPA.

4. Line 256-264: could you describe more about any of those 13 genes are potential candidates or consistent with any other published research? I understand that the flammable traits may not have may published results. Candidate for any traits highly correlated with it? You mentioned some in the discussion but you may want to explain more in the main text.

>>The supplemental Tables S5-S10 contain the gene descriptions and identification of the Arabidopsis orthologs for the candidate genes, differentially expressed genes and all expressed genes. The first version of this manuscript did include more speculation in the discussion and main text about flavonoid biosynthesis genes among the candidates, (a pathway we previously found to be relevant in brown cotton with natural flame retardance) but the previous reviewers at PLOS One suggested that we reduce that speculation.

5. Line 290: using highest or lowest HRC values instead of worst.

>>Done.

6. Line 327: remove “Found among”, and “were lines that”.

>>Done.

7. Line 335: remove “already”.

>>Done.

---

## [Decision Letter · Decision Letter 2]

22 Nov 2022

Flame resistant cotton lines generated by synergistic epistasis in a MAGIC population

PONE-D-22-19253R2

Dear Dr. Thyssen,

We’re pleased to inform you that your manuscript has been judged scientifically suitable for publication and will be formally accepted for publication once it meets all outstanding technical requirements.

Kind regards,

Yuan-Ming Zhang

Academic Editor

PLOS ONE

Additional Editor Comments (optional):

I have no further comments.

Reviewers' comments:

Reviewer's Responses to Questions

**Comments to the Author**

1. If the authors have adequately addressed your comments raised in a previous round of review and you feel that this manuscript is now acceptable for publication, you may indicate that here to bypass the “Comments to the Author” section, enter your conflict of interest statement in the “Confidential to Editor” section, and submit your "Accept" recommendation.

Reviewer #2: (No Response)

Reviewer #3: All comments have been addressed

2. Is the manuscript technically sound, and do the data support the conclusions?

Reviewer #2: (No Response)

Reviewer #3: Yes

3. Has the statistical analysis been performed appropriately and rigorously? 

Reviewer #2: (No Response)

Reviewer #3: Yes

4. Have the authors made all data underlying the findings in their manuscript fully available?

Reviewer #2: (No Response)

Reviewer #3: Yes

5. Is the manuscript presented in an intelligible fashion and written in standard English?

Reviewer #2: (No Response)

Reviewer #3: Yes

6. Review Comments to the Author

Reviewer #2: (No Response)

Reviewer #3: (No Response)

7. PLOS authors have the option to publish the peer review history of their article (what does this mean?). If published, this will include your full peer review and any attached files.

Reviewer #2: No

Reviewer #3: **Yes: **Shuyu Liu

---

## [Editor Report · Acceptance letter]

2 Dec 2022

PONE-D-22-19253R2 

Flame resistant cotton lines generated by synergistic epistasis in a MAGIC population 

Dear Dr. Thyssen:

I'm pleased to inform you that your manuscript has been deemed suitable for publication in PLOS ONE. Congratulations! Your manuscript is now with our production department. 

Kind regards, 

on behalf of

Dr. Yuan-Ming Zhang 

Academic Editor

PLOS ONE